# On Representing Extreme Experiences in Writing and Translation: Omid Tofighian on Translating the Manus Prison Narratives

Omid Tofighian [1,2]

1   School of the Arts and Media, University of New South Wales, Kensington, NSW 2052, Australia;
    omid_tofighian@yahoo.com
2   School of Law, Birkbeck, University of London, London WC1E 6DP, UK

**Abstract:** On 10 June 2021, the Norwegian translator Signe Prøis (for publisher Camino Forlag) organised an event with both Behrouz Boochani and Omid Tofighian (both by video link from New Zealand and Australia) in conversation with translation studies scholar Erlend Wichne (University of Agder, Norway; Agder forum for translation studies). The event was titled: 'Can I translate it? On representing extreme experiences in writing and translation'. The dialogue in this article features excerpts from the seminar with a focus on Tofighian's translation of Boochani's *No Friend but the Mountains: Writing from Manus* Prison (2018) into English. The topics covered include responsibility, translation as activism, some aspects of the broader context to translating *No Friend but the Mountains*, the role of place, and a shared philosophical activity.

**Keywords:** refugees; Behrouz Boochani; borders; exile; prison narratives; Australia

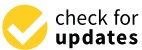



## 1. Introduction

Australia has built and manages immigration detention centres on islands in the Pacific thousands of kilometers away from the nation's international borders. Through the illegal practice (according to international law) of exiling and imprisoning people it deems undesirable and expendable, Australia reactivates and invigorates dimensions of its violent colonial legacy. The Australian border, therefore, remains fluid and extends into neighboring countries in unsettling, debilitating and irreparable ways. Exploitative agreements with poorer nations to construct prisons also enable Australia to impose on the sovereignty of those countries and disrupt and damage their socio-cultural, economic and moral fabric, thus maintaining a longstanding colonial dynamic. Australia was established as a penal colony by the British Empire; today, Australia transforms other islands into prisons.

Manus Island and the Republic of Nauru are recent examples of locations for Australia's immigration prison camps. The Australian government began exiling and incarcerating people seeking asylum in these offshore prisons between 2001 and 2008, which was referred to as the 'Pacific Solution'. Since 2012, the second iteration of the Pacific Solution, over four thousand people have been held in these carceral sites. Manus Island is part of Manus Province of Papua New Guinea (PNG): the detention centres on Manus have now been closed. Over one hundred men who were locked up in Manus for over six years are still being held in Port Moresby, the capital city of PNG, with no clear pathway to restarting their lives in a third country. Currently there are over one hundred men and women still being held in Nauru with no clear pathway to restarting their lives in a third country (the facilities in Manus and Port Moresby are for men traveling alone, and in Nauru for women, families and unaccompanied minors).

PNG is a former Australian colony, Nauru is a former protectorate (PNG gained independence in 1975, and Nauru in 1968). Since 2001 Australia has exploited these islands,

using them to warehouse people who travelled to Australia by boat to ask for protection. The natural environment of these locations has been destroyed in order to construct carceral sites of various forms for imprisoning and controlling people seeking asylum for an indefinite time period. The construction, management, security and maintenance contracts for the prison camps have mostly been given to multi-national companies; these extremely lucrative agreements, some of which were made after questionable tender processes, have enabled many companies to profit significantly from the misery of human beings. Approximately ten billion dollars has been spent since 2012 to sustain Australia's offshore detention industry. Australia's two major political parties are complicit; over time, numerous politicians and companies have devised even more complex and brutal technologies for controlling human movement. In conjunction with the mainstream media and a multitude of organizations, they have helped enhance the border-industrial complex, and they have contributed to further demonizing refugees and further securitizing and militarizing the border.

Border politics in Australia is a ruthless political and economic enterprise. Similar to most nation states around the world, border politics has become central to many debates in Australia regarding elections, national security, national interest and national identity. In 1992, the Australian government implemented the policy of mandatory and arbitrary detention designed to deter people seeking asylum by boat. In 2001, John Howard's Liberal government first introduced the Pacific Solution which, as mentioned above, involved the creation of offshore immigration detention centres. Stage two of the Pacific Solution was reintroduced by Julia Gillard's Labor government in 2012. Kevin Rudd took over leadership from Gillard in 2013, and soon after he introduced a new policy where anyone arriving by boat after 19 July 2013 would never be settled in Australia.

Behrouz Boochani fled Iran in 2013 after increasing persecution for his cultural and political advocacy for Kurdish rights and identity; a member of a marginalised and persecuted ethnic group, he was forced to leave his homeland soon after his journalist colleagues were arrested during a raid on their offices by Iranian authorities. Despite being rich in natural resources, the Kurdish regions in Iran are among the most impoverished in the country, and Kurds face systemic discrimination on all levels of society, culture and politics; this creates a deprived, humiliating and unbearable existence in Iran for Kurdish people like Boochani, similar to the situation faced by Kurds in other parts of Kurdistan (areas in Iraq, Syria and Turkey). Significantly, the recent nation-wide uprisings in Iran occurred after Kurdish-Iranian woman Zhina Mahsa Amini died from a brain injury after being violently detained by the state's 'morality police'. The now-iconic cry of resistance that has come to represent the struggle of Iranian people united against state violence, "*zan, zendegi, āzādi*" ("Woman, Life, Freedom"), is a translation of a resounding Kurdish political slogan.

After fleeing Iran and arriving in Indonesia, Boochani attempted to reach Australia by boat to ask for protection. He nearly drowned during his first attempt to travel to Australia when the boat sank. After he was returned to Indonesia, he embarked on a second journey. However, when the July 19 ruling was made, his boat was lost at sea for a week and, he arrived four days after the new law was announced. Like thousands of others fleeing dictatorship, discrimination and war he found himself trapped in Australia's border-industrial complex.

In 2013, Australia experienced a change in government with the Liberal-National Coalition defeating the Labor Party. The new Prime Minister, Liberal leader Tony Abbott, continued to further militarize Australia's borders with the support of Scott Morrison as the new immigration minister. With Morrison as immigration minister, the border-industrial complex entered a new phase of brutality: Operation Sovereign Borders (Peter Dutton took over from Morrison as immigration minister in 2014). To date, this regime has caused the deaths of 14 people who were held in Manus and Nauru (not including those who lost their lives after deportation). Morrison became Australia's Prime Minister in 2018 and remained in power until 2022, with Dutton taking over the Liberal Party leadership after years as the Minister for Home Affairs; the Department of Home Affairs is a newly-created 'super-

department' (with Michael Pezzullo as secretary). Operation Sovereign Borders remains in operation following the Labor Party win in the 2022 federal election, with current Prime Minister Anthony Albanese. Australia's border regime is in breach of international law. People seeking asylum by boat are exercising their right to seek asylum under international law; the people who have been held in Australia's immigration detention system have been incarcerated without ever committing a crime and without trial. Behrouz's writing and other creative forms of resistance while detained documented this tragedy and humanitarian crisis. His critical analysis from inside the prison created a new discourse for exposing border violence. Together with his translator and collaborator Omid Tofighian—a Sydney-based academic and activist who first made contact via Facebook and WhatsApp when Boochani was still using a smuggled mobile phone—they created Manus Prison Theory, a multidimensional theoretical and creative framework for understanding the recent incarceration of refugees on islands. It is a critical perspective that situates immigration detention within a long history of oppression, domination and subjugation which began with the dispossession and genocide of First Nations peoples and the establishment of the land as a penal colony for the British Empire.

Manus Prison Theory is an evolving ecosystem of philosophical and artistic initiatives coupled with political action and community advocacy; it emerged out of over six years of collaboration between Boochani and Tofighian. It can be described as a collective form of knowledge production that continually incorporates new collaborators, which Tofighian has referred to as a 'shared philosophical activity' (Tofighian 2018b). This growing body of knowledge involves devising effective political actions aimed at abolishing Australia's carceral-border logic and transforming the nation's dominant social imaginary using diverse strategies and techniques; this work has matured into original and radical networks of scholarship, art and collective action. For instance, Boochani and Tofighian assert that the border-industrial complex is global and is interlinked with many different forms of intersectional discrimination, exploitation and subjugation. They argue that the systems of oppression that characterize the border-industrial complex are interconnected, mutually-reinforcing and self-replicating. As such, the detention industry is indispensably connected to other forms of violence, especially oppression in contemporary Australian society; in relation to Australian imperialism in the Pacific and beyond; and from Australia's colonial past. Significantly, Manus Prison Theory acknowledges and builds on diverse decolonial and intersectional perspectives and practices; as a philosophy of resistance, some of its contributions involve the use of concepts such as kyriarchy (a term from radical feminist theology first introduced by Elisabeth Schüssler Fiorenza); Boochani's incorporation of Kurdish resistance and history; and the introduction of what Tofighian refers to as 'horrific surrealism' (Tofighian 2018b, 2021). In sum, Manus Prison Theory is dedicated to examining and challenging Australian border violence by identifying it as a dimension of the nation state's dominant social imaginary and also by critiquing the symmetrical relationship between the Australian border and Australia's socio-political structures and institutions (Tofighian 2020).

Boochani started collaborating with Tofighian on different projects at the beginning of 2016. Together, they began strategizing ways to expose and dismantle Australia's border-industrial complex, and one of their aims was to radically transform the mainstream image of refugees. They attempted to disrupt the social imaginary in Australia pertaining to displaced and exiled peoples. The social imaginary—or rather, a colonial imaginary when considering the dominant role of Australia's colonial legacy in representing and determining domestic and international relationships, interactions and futures—involves the material, symbolic and epistemic conditions that render displaced and exiled peoples weak and without agency. Many widely-held assumptions about refugees are often patronizing and debilitating; they are determining factors that result in refugees being subject to inhumane punishment and abject conditions. That is, the social/colonial imaginary pertaining to refugees functions interdependently with other forms of bordering and therefore must be considered a fundamental component of the border violence nexus. As part of Manus

Prison Theory, Boochani and Tofighian propose new acts of debordering. Tofighian's support primarily involved editing and translating Boochani's journalism, speeches and statements from Persian/Farsi to English; co-authoring scholarship; creating subtitles for his film *Chauka, Please Tell Us the Time* (Boochani and Kamali Sarvestani 2017); assuming the role of translator and editor of his multi-award-winning and genre-defying autobiographical novel *No Friend but the Mountains: Writing from Manus Prison* (Boochani 2018); and co-translating and co-editing (with Moones Mansoubi) the collection *Freedom, Only Freedom: The Prison Writings of Behrouz Boochani* (Boochani, forthcoming, edited by Tofighian and Mansoubi).

*No Friend but the Mountains* was written by Boochani during the first five years of his incarceration in the Australian-run offshore immigration detention centre on Manus Island (the original site within the Lombrum Naval Base). It was written completely on WhatsApp via hundreds of text messages and sent to friends and colleagues in Australia for translation, editing and publication (Farsi/Persian-English). Moones Mansoubi—Boochani's first translator and collaborator—collated most of the messages into individual chapters and created PDFs for Tofighian to translate; Boochani continued to send text messages to Tofighian to insert/change the text, with the final chapter arriving directly to Tofighian via one text. The final chapter (Chapter 12: 'In Twilight/The Colours of War') was completed during the 23-day siege and forced removal of refugees to new prison camps on Manus in October–November 2017; during this period, the Australian and PNG authorities discontinued food, water, power, medication and services and withdrew security and management personnel in order to force the refugees to move to the new camps. After 23 days of resistance, the PNG police and military were sent into the prison camp to brutally transfer the detainees.

Soon after publication, *No Friend but the Mountains* was awarded the 2019 Victorian Prize for Literature, among many other prestigious awards, and Boochani (through video link from the prison until Nov 2019, and afterwards from New Zealand) was invited to speak at many Australian and international festivals, book launches, seminars, campaign events and conferences, together with his translator and collaborator Tofighian (mostly in person). Since the release of the book, a few of these events have focused on translation.

On 10 June 2021, the Norwegian translator Signe Prøis (for publisher Camino Forlag) organised an event with both Boochani and Tofighian (both by video link from New Zealand and Australia) in conversation with translation studies scholar Erlend Wichne (University of Agder, Norway; Agder Forum for translation studies). The event was conducted in English and was titled: 'Can I translate it? On representing extreme experiences in writing and translation'. This event was distinct due to its focus on writing, translation, collaboration and publishing as resistance; Manus Prison as a site of knowledge production; and the forms of shared knowledge produced between author and translator.

The following dialogue, 'On Representing Extreme Experiences in Writing and Translation: Omid Tofighian on Translating the Manus Prison Narratives', is from this seminar. The excerpts chosen for this article address Tofighian's translation of the book into English and the collaborative shared philosophical activity necessary for producing *No Friend but the Mountains* and similar works. The topics covered included responsibility and translation as activism, some aspects of the broader context to translating *No Friend but the Mountains*, place and a shared philosophical activity (see Supplementary Materials with a reading list of articles by Tofighian and Boochani about related themes and issues).

## 2. The House of Literature in Fredrikstad, Norway 2021

**Erlend:** In your preface to the book *No Friend but the Mountains*, you write that to translate the voice and perspective in Boochani's texts, one needs a big imagination (Tofighian 2018a). I imagine that for you there must have been a point in time when you were realizing that you were going to translate and make a book of Boochani's texts sent out from Manus prison. I am wondering about when and how you made this realization, and I am also wondering how you evaluated your own situation in contrast to that of Boochani's, where you, as a philosopher and translator, were working from within the

Australian frontiers that Boochani had tried to enter and that were depriving him and his fellow prisoners from expressing their abilities as human beings. What could Boochani see of the world that you could not see (Tofighian 2018a)? And what kind of responsibility did you recognize when taking the role as translator for his accounts?

**Omid:** Thank you for that question. This is something that, to be honest, I have been thinking about from day one, and it is something that has grown and developed from the time I started working with Behrouz. And still today, I continue to learn so many new things after reflecting on the whole process, and seeing the directions that the book has moved in and the way it has been taken up. And now we are talking to people in Norway about it, and about the translation of a translation (all translations of the book are based on the English translation). This is a really fascinating topic and I think a whole book needs to be written about this particular interaction. *No Friend but the Mountains* needs to be understood as one moment, one phase, one example of a very long process. And that process began before the book. Originally I was translating Behrouz's journalism and working on different kinds of activism with Behrouz; we were thinking about ways to challenge the system, dismantle the system, thinking about abolition in this context, not just in terms of borders and immigration detention, but much wider than that. This developed, it came together and influenced the book. After we started working on the book—and then, after the book was finished—this kind of thinking and acting continued and continues still today. The book is actually part of a much larger story and is not the main focus; the book is one element in a very complex, multidimensional narrative that includes many other people. And I think here it is important to mention how I came across Behrouz's work; it was through another translator. Again, the importance of translation, this shows why translation is so important. I came across Behrouz's work and him as a person through the translations of his first translator, Moones Mansoubi. She lives in Sydney and was working with Behrouz for quite some time before I met him; she translated many articles for him. It was through his first article that was published in *The Guardian*—and the first time that he published under his real name—that I came across his situation and his writing, and then I got in touch and we started a relationship. Soon after that I began translating for him, amongst other things, and then that led to the opportunity to translate the book.

It is interesting to examine the notion of place; we are addressing a particular place here that I call a neo-colonial experiment. It is a whole new phenomenon. I am glad Behrouz explains that it needs to be understood in relation to this age, this time, because we are talking about a form of illegal imprisonment that is being conducted by a liberal democracy. And one of the key features is the paradox associated with a liberal democracy creating a particularly brutal space that it controls and manages, a space that Australia is indispensably connected to and which resembles tactics, techniques, ideologies we often associate with fascism. One of the key tropes in this book is paradox. And I think Behrouz worked paradox really well into his book because of the fact that he was engulfed in paradox in the prison camp. This paradox also pertains to the political situation, the border politics in relation to Australia. So here we have a prison camp where people have not been through a court system, have not been convicted of any crime, but are being kept indefinitely. Unlike a prison where you know your sentence, in immigration detention you are being kept indefinitely. At a certain point, phones were illegal, but then they became legal at a certain point; they were no longer criminalised, and there was no ban on phones. At first, people were also locked up in smaller prison camps inside the larger prison, so they could not even communicate with each other. But then at a certain point, the doors opened and people could leave the prison camp, and then the whole island became a prison. So what we are talking about here is maybe something unprecedented in history. And I think paradox or even surrealism is one of the best ways to think about it; I introduced the term 'horrific surrealism'. The last thing that I will mention in terms of translation and the importance of translation and interpreting the whole phenomenon, including the book, is that unlike some of the writers who were imprisoned by fascist regimes in Europe who are looking to European tradition, European philosophy, European poetry, European

arts in terms of their salvation, in terms of their release and resistance, Behrouz here was not looking to the Western tradition or to the Australian cultural scene and its history; he was going back to his Kurdish heritage. Not only that, he was going back to a Kurdish heritage that was particular to his own region, particular to his own village. And I think this is really striking. There is so much to say about it. He connected with the Indigenous people on Manus Island and drew strength from their history and from their stance against colonialism. So as a translator, I had to do so much work to think about how all of these different pieces fit together and how Behrouz's voice comes out of the intersection between all of these facets.

**Erlend:** I have a question for you about language. Many people work with or within the Australian immigration prison system, and among them we find interpreters. In general, there is a very pessimistic development in the image of the interpreter in this book. First, Behrouz imagines a Kurdish interpreter seeing her own story in him and in his physical presence, almost; this is when he enters the plane from Christmas Island to Manus. Then later, the group of interpreters are perceived as under a total control of the kyriarchal system. It is written that in many ways, the interpreters are the most helpless in the prison; they are deprived of their identity, they are like thinking loud speakers, and they are under total control and deprived of their ability to express sympathy. One would think that the linguistic competence that they have was going to serve more noble goals than enabling the Australian kyriarchy to oppress its prisoners in a language that they understand. But in a way, the interpreters have lost their language. I would say not by not speaking it, but by being forced to turn it against the people in which they see themselves and their own personal history. And your activity as a translator in principle is not that different from that of the interpreters. Why is your agency as a translator so much wider than the agency of the interpreters in this system?

**Omid:** Behrouz says that knowing the system helped him survive, so really that intellectual activity, that deep analysis, that sophisticated critique, was basically some of the reasons why he was able to come out of the system with his faculties, with his mind in a good place. He felt healthy coming out of the system, even though the system was designed to take his identity and ruin him psychologically, emotionally and spiritually, as well as physically. But I think this particular point that Behrouz makes relates directly to what you are asking about in relation to interpreters and translators. This is just my own personal interpretation and I think it will be interesting to do research on this topic, but I think a lot of the people working in the system as interpreters, as translators, as caseworkers, as counselors, as doctors, I think some of them actually feel, at least in the beginning or before going into the system, that they are doing a good thing. Some of them think that they are actually there to help the people who are imprisoned. And I think they have this imagination that if they did their work really well, if they really treated the detainees with respect, if they were attentive to all of their needs and perform their duties in exceptional ways, then they could help them find freedom very quickly and there would be no need to detain people in this particular way for such a long time. I think genuinely some people working in the system feel that they can help, they can do something positive. But I think this goes back to Behrouz's point about knowing the system. I think right across Australian society and politics a lot of people do not understand the system. Behrouz makes a really great point in his book and in other places that even people in the system, those who designed the system, even they do not know the system. The system is so complex, it is so fluid, it is organic, it grows, it morphs, it multiplies, it changes direction, it defends itself, it justifies itself in really obscure ways, in really unpredictable ways. So I think a lot of the people working in the system who are there because they think they can make a change, they in fact do not know the system. I am not saying that I know the system. I am not saying that I have cracked some kind of code, or anything like that. But we have to be clear on a number of things. I worked with Behrouz on a day-to-day basis, as he has said, before we were in conversation every day, analyzing, talking, sharing ideas. I was going away and researching so much of what he was telling me. I was contributing

a lot of my own ideas as well and sharing a lot, and we transformed and changed a lot of things as well. It is impossible that when you spend this much time thinking about a system that you do not eventually go into the depths or into the belly of that system in some way. You know, I felt at some point that I was walking through the prison. I was in the prison for a certain period of time when we were working so closely. I could feel the anxiety, the stress, especially when Behrouz would send me works to translate—his journalism, for instance—and it had to be done the next day. It had to be done that very day sometimes. This remains in your body, I actually still have that in my body now. I mean, those years of doing that kind of work is in my body. It will take me years to get it out of my system. By doing this kind of work in such an extreme way you come to know the system, you come to understand something particular about state-sanctioned violence. You know something particular about border regimes in Australia after doing this for such a long time. And that is why I took a completely different approach. My approach was not just political, not just practical, but it was also epistemic, it was symbolic, and it was about attacking the colonial imaginary. So I was taking a very different approach to changing things; we just had two telephones, two people, and one book. We made a rupture, we injured the system in a way that I think people who have spent many, many years trying to change it were not able to. The last point that I will make is about my own experience as a marginalized person in Iran and someone who lives in exile and who has also experienced racism in Australia, someone who has lived in Australia and researches and is committed to decolonisation and the study of different aspects of settler colonialism here. That gave me a good platform to work with someone like Behrouz. This helped me to know the system. So when Behrouz talks about colonialism, it is not something unfamiliar to me; when he talks about discrimination and the kyriarchal system, it was, in fact, something that I was seeing around me quite regularly.

**Erlend:** I want to continue talking about the connection between experience and language. We have been speaking about creativity, intellectual work and theorizing, and also loss of language in the case of the interpreters. I think that we are starting to see the importance of language when giving meaning to extreme experiences like the ones described in *No Friend but the Mountains*. And these invite us to consider the possibility of translating texts written by people who have experienced this by people that do not necessarily have the exact same experiences. And this issue is not unique for the translation of refugee experience, it's probably a case of all translation that you do not have exactly the same experiences. But in cases like refugee experiences and also gendered and racialized experience, this might be relevant to discuss within a framework like this, where it is evident that language is formed by specific experiences. And in Europe and in Norway, this has gotten an expression very recently in the debates over who might or who might not translate the poetry written by Amanda Gorman, for example. I am interested to hear your perspective on this issue. How do you perceive the issue of translating writings about experiences which are not yours?

**Omid:** Thanks for raising this issue. I have to admit, I have only very briefly looked into the debate around the translation of Amanda Goodman's work. I know a little bit about it, but I have to look into it a lot further. But to respond to your question, maybe I will try to put a different angle on it. Rather than limit the debate about who is able to translate or whether it will be a good translation, whether it will be accurate, whether preference should be given to people who have the same kind of identity or socio-cultural position, instead maybe we could talk about the conditions that actually give rise to the possibilities of a translation. For instance, I am not trained as a translator, and even today I kind of find it weird to think of myself as a translator. I came into translation by accident. I started translating for people a number of years ago, maybe eight years ago, and that was only just a few lines here and there. When I met Behrouz, I had never really translated a full article or a complete text. But I did grow up interpreting for my parents since I was a child. So I was already thinking in two languages. But still, that was very different from translating a text. So, my first real substantial experience in translation came from working

with Behrouz, and it came from an activist space. When I say the conditions for translation, I think it is important to look at the political dimensions regarding: who is encouraged to learn translation?; to appreciate translation?; why isn't translation being taught in some spaces?; how is it being taught?; who is being encouraged to engage in translation? I think these questions about who should be translating someone's work and who shouldn't be may not even be a major debate if we already had a long tradition, and already had a political vision about what translation means in our lives, what it means in our world, and what kind of work should be done as a result of teaching translation. Why is it that there are not more people with backgrounds in displacement and exile being encouraged, funded and employed to do translation work that would end up resulting in something like *No Friend but the Mountains*? Why is it that now, after winning all of these prizes and all of this international attention and all of this different research being done, it is still almost impossible for me to get support for my next translation project with Behrouz? Why is that the case? For me it is surprising, and it is unsurprising at the same time. Why is this the case? Why are there not more people with lived experience of particular situations, particular kinds of struggles, involved in his work? Why are there no programmes? Why are there no benefactors available to really think about the importance of this to everyone's knowledge, everyone's understanding of the world, everyone's understanding of each other? Maybe thinking about this debate in those terms might add something significant and take the conversation in other directions, maybe more important directions.

**Erlend:** I think in your essays published with Behrouz's book, you introduce a very important concept of translation which could be useful to what you outline; exactly the idea of translation as a shared philosophical activity. I think that is a beautiful concept which you write clearly about in your translator's note and which gives a very good expression of what translation can be and how it can be more a dialogic than, sort of, a monologic process from the translators. Could you say something about this shared philosophical activity regarding translation?

**Omid:** If we want to think about it a lot more radically, I also consider Behrouz as one of the translators. We were always in conversation every day, always clarifying terms, I was always explaining the kinds of experiments that I was engaging in. Behrouz was giving feedback on the translation, he was helping with the translation, he is part of that shared philosophical activity. Moones Mansoubi was my translation consultant and Behrouz's first translator, and another translation consultant was Sajad Kabani, an Iranian researcher who was in Sydney at that time. Consider the kind of critiques that are made about this idea of the single, lone genius who creates theories and philosophies and writes books and articles; we need to move away from that idea. We are all relational. We all interact with each other, and particularly in these important projects, these really pivotal, influential projects that end up having a huge impact on different fields and on different people from different places. It is more the case that relationships make the product, relationships make the outcome and something like *No Friend but the Mountains* could never have been produced if it were not for people who were all committed, all focused on making it happen and all investing and believing in what Behrouz was writing, investing in Behrouz's resistance, really seeing the outcome. I realized that this book was going to be a masterpiece after reading two or three pages; I really believed in it from that point, I said, "This is going to be something special". I can see something, I can feel something here, I can see how this can be used to make change, how it can be leveraged to make real transformation, to change the way people see displacement, exile and incarceration. I could see how we could make a rupture in the system. Once this team came together, everyone had the right kind of vision and the right kind of commitment. We had many bodies, but one mind. In terms of philosophy of mind or in terms of metaphysics, I think there are really important discussions to have about this phenomenon. You have situations where there is one body, many identities. But what we had in our case was really unique because we had many bodies, one identity. And it was only when we were working on the project that this took place, that this was solidified. Now we are all separate and we are all moving on with our

lives, doing different things. But that moment, that special moment, I think that is one of the magical aspects of *No Friend but the Mountains*. And imagine if we could invest more, if we could create the conditions again, going back to my point about creating the right conditions for a translation. A translation that involves so many different stars aligning, so many different kinds of principles and beliefs, notions and commitments for change, all coming together. If we could make that happen again, if we could replicate that shared philosophical activity, not just in this space but in other spaces, I think we will see other magical moments, other special outcomes.

**Supplementary Materials:** The following supporting information can be downloaded at: https://www.mdpi.com/article/10.3390/h11060141/s1, Selection of Boochani's and Tofighian's Writings (in Addition to Some Collaborators).

**Funding:** This research received no external funding.

**Institutional Review Board Statement:** Not applicable.

**Informed Consent Statement:** Not applicable.

**Data Availability Statement:** Not applicable.

**Conflicts of Interest:** The author declares no conflict of interest.

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
