# Peer review of "On Representing Extreme Experiences in Writing and Translation: Omid Tofighian on Translating the Manus Prison Narratives"

_humanities, doi:10.3390/h11060141_

Round 1

Reviewer 1 Report

The main contribution of the piece is the transcription of an interview related to the collaborative translation process of the work No Friend but the Mountains. The interview raises important questions related to ethics and translation, as well as the relationship between translation and efforts to decolonize a global border regime/ border-industrial complex.

The commentary that precedes the interview transcription offers useful context to the non-specialist about the author of No Friend but the Mountains, as well as the history of border policies and migrant detention practices in Australia. It is here that some further clarification would be useful to the reader. Points to explain further include:

why did Boochani flee Iran in 2013?

p. 2, the first full paragraph I feel could be divided into two paragraphs. The topic seems to shift abruptly from Boochani's personal experience to the way distinct governments in Australia made decisions related to border/ deportation policy. It seems a paragraph break (with an appropriate transition) should come between the sentences (line 50) that end "the new law was announced" and "In 2013 Australia"

on p. 2, line 64, the author writes that "people seeking asylum by boat are exercising their rights"--what rights are referred to here? the right to seek asylum?

p. 2, line 68, it is unclear whether the collaboration with Omid Tofighian occurred within the prison, or whether Tofighian was outside of prison

p. 2, line 79, there is a quick renaming of "social imaginary" as "colonial imaginary" that could be explained a bit further at this point. The author shows the connection a few sentences later, but it would be helpful to clarify how the social imaginary is a colonial imaginary when it is first mentioned.

p. 2, line 87, there is mention of translating Boochani's work--state from what language to what language (I assume from Persian/ Farsi to English, but is this correct?)

p. 3, line 98, "Mansoubi collated most of the messages ..." who is Mansoubi?  

p. 3, line 102, which siege? Who are the parties involved here?

p. 3, line 105, Boochani spoke through video link, I assume from prison?

p. 3, line 111, specify that University of Agder is in Norway)

p. 3, line 114, was the conversation /dialogue held in English? I assume so, since there is no mention of a translator of the dialogue, but I don't want to assume

I also wonder whether there is a grammatical problem on p. 1, line 28--should it read "using them to warehouse people whom it deems undesirable" instead of "who it deems undesirable"?

Generally, this is a very useful and interesting piece on the ethics of translating in the context of a particular border regime. The bibliography of Boochani and Tofighian's writings is extremely useful.

I would suggest re-reading the commentary preceding the dialogue while keeping in mind the perspective of someone who is not familiar with that particular region of the world, or with Boochani and Tofighian, and editing to clarify and elaborate terms that might not be known.

There seem to be a number of typos, particularly in the transcription of the interview--for example, missing words or extra phrases.

Author Response

Thank you for your helpful and important comments. I appreciate your thought and effort and have updated my contribution. Please see some brief responses to your feedback below.

I expanded important areas of the introductory commentry with particular focus on the points you highlighted as important such as the border-industrial complex, history, the author and translation.

Also, I explained more about why Boochani fled Iran in 2013 by highlighting the Kurdish struggle.

On p. 2 I divided the first full paragraph into two paragraphs as you suggested and added an appropriate transition. I also clarified what I meant by international right to seek asylum. I expained that the collaboration began with Behrouz inside the prison using a smuggled phone and me based in Sydney using Facebook and WhatsApp.

I clarified the quick renaming of "social imaginary" as "colonial imaginary". I indicated that the translation of Boochani's work is from Persian/ Farsi to English.

I mention who Mansoubi is and her role in the process. I explain some details of the siege and why it is important. I mention that Behrouz was using video link from prison and then from NZ.

I specify the host's university being in Norway and that the conversation /dialogue was held in English.

In general, I re-edited and updated some important information.

Best wishes.

Reviewer 2 Report

There is no clear philosophy behind the “excerpts from the seminar.” Why these sections? and why this specific seminar? They have participated in several other seminars.

In p.2 of 9, I would like to see a brief introduction to the situation of the Kurds in general and the Kurds in Iran in particular. 

In p.2 of 9, the writer states: "Together with his translator and collaborator Omid Tofighian they created Manus Prison Theory." I would like to see more about "Manus Prison Theory." 

Author Response

Thank you for your helpful and important comments. I appreciate your thought and effort and have updated my contribution. Please see some brief responses to your feedback below.

I give a short explanation for why the seminar is unique and the reasoning behind the excerpts from the seminar. I indicate that it was different to most other seminars.

I give details about how the Kurdish struggle relates to Boochani fleeing Iran and mention the situation of the Kurds in general; I indicate why this is important to consider now in respect to recent uprisings in Iran. 

I also give a good account of Manus Prison Theory, how we created it together and why.

Best wishes.